# An Asymmetric Analysis of the Influence That Economic Policy Uncertainty, Institutional Quality, and Corruption Level Have on India's Digital Banking Services and Banking Stability

Aamir Aijaz Syed [1], Muhammad Abdul Kamal [2], Assad Ullah [3] and Simon Grima [4,*]

1 Institute of Management, Commerce and Economics, Shri Ramswaroop Memorial University, Lucknow 226016, India; aamirank@gmail.com
2 Department of Economics, Abdul Wali Khan University, Mardan 23200, Pakistan; kamal@awkum.edu.pk
3 Department of Economics, Henan University, Kaifeng 475001, China; assad@henu.edu.cn
4 Department of Insurance and Risk Management, Faculty of Economics Management and Accountancy, University of Malta, MSD 2080 Msida, Malta
* Correspondence: simon.grima@um.edu.mt

**Abstract:** Motivated by the unprecedented high levels of recent economic policy uncertainty, the current study examines the influence of economic policy uncertainty, institutional quality, and corruption level on the Indian banking stability and the growth of digital financial services. Using the Baker et al.'s economic policy uncertainty index and nonlinear autoregressive distribution lag model on the data set of banking variables from 2004 to 2019, we infer the following findings. The unit root and the structural break tests confirm the presence of structural breaks and mixed order of integrations. Besides, the long-run nonlinear autoregressive distribution lag results substantiate a long-run asymmetric relationship between the explanatory variables (economic policy uncertainty, institutional quality, corruption level) and the outcome variables (digital banking services and banking stability). The study reveals that a 1 percent increase in the economic policy uncertainty increases nonperforming loans (proxy to measure banking stability) by 1.48 percent and decreases Z-score (proxy to measure banking stability) by −1.12 percent. Likewise, a 1 percent increase in policy uncertainty reduces the progress of digital financial services by −1.23 percent in India. In addition, the study also depicts a long-run cointegration between the explanatory and the outcome variables. Overall, the study shows significant evidence that policy uncertainty, corruption, and institutional regulation hampers Indian banking stability and digital growth. The study offers several policy implications to understand the adverse effects of economic policy uncertainty on the Indian banking sector.

**Keywords:** EPU; corruption; institutional regulation; NPLs; digital growth



## 1. Introduction

Over the years, the financial crisis has garnered the attention of researchers to investigate the influence of macroeconomic instabilities on the countries' economic and banking structures [1]. Especially after the global financial crisis, several tail events have influenced the business and economic environment. In the quest to mitigate the severe shocks of the financial crisis, countries have taken drastic measures to strengthen their banking and economic fundamentals. For instance, tightening regulatory norms, capital buffers, reinforcement of fiscal stimulus, structural adjustments, institutional regulations, etc. [2,3]. These stringent regulatory norms and economic restructuring have created uncertainties in the monetary and fiscal policies, further affecting the government directions and inversely influencing the business prospects [4]. The inability to properly implement regulatory measures and slow policy inactions have negatively influenced the employment and investment cycles. During the recent COVID-19 pandemic, economic disturbances and sluggish

growth have further exacerbated the economic uncertainties. These cultivating uncertainties have attracted the attention of researchers to evaluate the negative consequences of economic policy uncertainties (EPU) on the economic and financial aspects [5,6]. These curiosities have led to the invention of new indices to measure time-varying economic policy uncertainties [4]. Previous studies have concluded that uncertainties in government policies have a magnifying impact on the real economy [7]. For instance, it influences the investment cycles, restricts corporate profits, hampers employment opportunities, and reduces saving rates [8,9]. In addition, the financial sector, especially banks, is also significantly affected by these economic uncertainties. Such kinds of policy uncertainties not only hamper banking profitability and stability but also restrict the progress of financial inclusion. Altunbas et al. [10] highlighted that the global economic uncertainties and the consequences of financial recession has raised serious apprehensions about banking stability and efficiency. The recent COVID-19 pandemic has further worsened the situation. It has raised serious doubts about how the banks will overcome these uncertainties, particularly in the case of emerging countries, where banks are already reeling under high non-performing loans (NPLs) and low profitability [11]. In addition, few recent studies also concluded that economic policy uncertainty gauges the level of uncertainty pertaining to institutional quality. Policy uncertainties weaken institutional regulation and increase corruption, thus promoting the chances of banking instability in the economy [12].

Against this backdrop, this current study examines the influence of institutional variables (EPU, institutional regulation, and corruption levels) on Indian banking stability and the growth of digital banking services. The present study is the first attempt to examine the above relationship for an emerging economy. Most studies have individually measured the consequences of these institutional variables on banking stability [13]. However, in the present study, we cumulatively investigate the effect of these institutional variables on banking stability. We have constituted all the variables together because they represent a similar class of institutional variables and analyzing them together will provide more robust estimates. Moreover, a few studies have simultaneously examined the influence of EPU on banking stability and financial inclusion [14]. However, the present study contributes to the extant literature by individually analyzing the effect of EPU on banking stability and digital financial services. We would like to independently examine the consequences of higher EPU on banking stability and digital financial services because emerging countries such as India still have a low percentage of digital growth compared to developed countries [15,16]. Moreover, the outreach of digital financial services and banking stability are also two prominent issues in emerging countries. The total NPLs of Indian banks are more than 9 percent, which is more than the global average of 5.5 percent. Likewise, India also ranks low in terms of the distribution of digital financial services compared to other developed and emerging countries. Therefore, analyzing both variables individually will provide more valuable insight for emerging countries. So, through interaction analysis in future studies, we can examine the consequences of EPU and digital growth on banking stability. The present study also contributes toward extant literature by including two proxies for measuring banking stability, which will provide more comprehensive results. Because in most of the extant literature, only Z-score is included as a proxy for measuring banking stability [16]. Finally, the present study also adds toward extant literature by including other institutional and control variables in the empirical analysis which were not covered in the previous literature.

The underlying explanation strengthens the reason for including India in our empirical analysis. First, India is among the fastest emerging economies, endowed with the highest middle-age population in Asia, providing a competitive edge compared to other countries. However, countries with huge populations are also more susceptible to economic disturbances during economic uncertainties. Second, India has a market-oriented approach and economic integration with almost all the developed and emerging economies. Therefore, economic uncertainties may have a significant spillover influence on its economic policies [17]. Third, India has a well-developed and emerging financial market, attracting

capital investment from developed countries. For instance, India has received the highest ever FDI inflow in 2020–2021. It surged by 10 percent to USD 81.72 billion, and the FDI during May 2021 was USD 12.1 billion, which is 203 percent higher than May 2020. Therefore, any disturbances in its economic policy may impact foreign investment, interest, and exchange rate volatility. Fourth, India relies heavily on its banking industry, which contributes to more than 8 percent of India's GDP. The Indian banking industry comprises a large cluster of public, private, and foreign banks. A large number of public, private, and foreign banks results in fragmented banking, which leads to the dissemination of funds into a large number of small banks, thus resulting in a weak banking structure [17]. In addition, the banking condition of some of India's public sector banks is not satisfactory. The public sector bank of India is facing the problem of higher NPLs, low financial intermediation, and banking profitability [18]. For instance, the cumulative percentage of NPLs is 9 percent, which is higher than the global average. Therefore, higher integration from the developed countries and internal and external economic uncertainty may impact the banking industry. Finally, the global focus is also shifting toward India because of the increased share in international trade and growing competitive powers. Considering the above reasons, we have considered India as a case study for empirical investigation.

The novelty of this paper relies on the following aspects. To begin with, this is the first study, as per the authors' knowledge, which investigates the influence of economic policy uncertainty on the Indian banking stability and digital financial services. The rationale for examining the effect of EPU on banking stability in India has already been outlined. However, digital financial services are also included in our study because emerging countries such as India have a low percentage of financial inclusion compared to developed countries. For instance, according to the global finder World Bank report, globally, India ranks 49th in terms of the share of the population with access to financial services. Previous studies have concluded that the growth in digital financial services enhances financial inclusion and contributes toward banking sector stability. We can collaborate this from the case of developed countries [19]. Therefore, we have also tried to investigate how economic policy uncertainties hamper the progress of digital financial services, especially in emerging countries. To measure the influence of digital growth, we have used the percentage of mobile money transactions. It is one of the widely used proxies to measure the dissemination of digital financial services. Furthermore, we have also included institutional regulation and corruption levels in our empirical analysis because all three variables constitute institutional variables and higher EPU entails weak institutional regulation and high corruption levels. In addition, to measure the corruption level, unlike previous studies, we have employed the newly developed corruption index of the International Country Risk Guide (ICRG), which is more robust and precisely captures the financial corruption level in the economy. Hence, this will add to the existing literature on the nexus between EPU, institutional regulation, corruption level, banking stability, and digital financial services. In addition, to measure the above relationship, we have also applied the nonlinear autoregressive distribution lag (NARDL) model together with advanced structural break and unit root test to add to the methodological novelty of the current study. The nonlinear autoregressive approach allows for modeling simultaneously asymmetric nonlinearity and cointegration among the underlying variables in a single equation framework. Another prime advantage of the NARDL model is its flexibility as this approach does not require all the variables to have the same order of integration, that is, the variables can be integrated of order one or not integrated. In addition, this framework permits testing for hidden cointegration and to differentiate among linear cointegration, nonlinear cointegration, and lack of cointegration. Two variables may not exhibit cointegration, but there is the possibility that their positive negative component may move together in the long run. Finally, to make it more comprehensive and novel, we have included two proxies for measuring banking stability (Z-score and NPLs), unlike previous literature. Although, there are several proxies to measure the banking sector's stability, for example, total capital ratio or leverage. However, Z-score and NPLs are the most widely used in the empirical banking literature to reflect

banking solvency and stability. It is also one of the indicators used by the World Bank in their Global Financial Development Database to measure financial institutions' soundness. Therefore, considering the importance of the above proxies, we have included them in our empirical analysis.

The rest of the study proceeds as follows: Section 2 presents the theoretical framework and literature review. Section 3 discusses the variable and methodology used in the empirical analysis. Section 4 highlights the detailed results analysis and interpretation. Finally, the last section covers the concluding remarks and policy implications.

## 2. Review of Literature

Since the global financial recession of 2007–2008, economists have tried to maintain stability in the economic policies to regain financial sector stability. A stable economic environment implies fewer negative macroeconomic shocks to an economy's functioning. Besides, it also promotes the overall development of all the prominent industries. Several previous studies have concluded that the complex nature of the economic environment creates uncertainties in the economic policies and affects the macroeconomic fundamentals [20–22]. However, there are still limited studies on the interaction between EPU and the banking function and stability. The subsequent section provides a comprehensive summary of the nexus between EPU, banking stability, and digital financial services.

### 2.1. Empirical Literature Review on the Nexus of EPU, Banking Stability, and Digital Financial Services

Nguyen [13] conducted one of the recent studies on the nexus between EPU and banking stability. The study concluded that in 900 commercial banks in 8 major European countries, EPU has a significant impact on banking stability. However, banking regulations and supervision provide a cushion to the banking stability during economic policy uncertainty. Bernal et al. [23] advocated that EPU influences bank functioning through the mediation role of the government deficit, output loss, and cash volatility. The adverse economic conditions worsen economic fundamentals, which eventually impact corporate profits and, thus, increase credit default and NPLs. The study conducted by Francis et al. [24] also supported the above findings. In addition, the study also concluded that economic policy uncertainty hampers corporate investment decisions and lowers banking profitability. Policy uncertainties create a pessimistic business environment, and firms are reluctant to make new long-term investment decisions. Since long-term investment ventures are expensive to reverse and workers are costly to employ and discharge, businesses seek for more information and scale back their commitments during times of high policy uncertainty, a phenomenon known as the "caution effect". The study conducted by Chi and Li [25] highlighted that at the macro level, higher EPU creates reduced aggregate investment, business opportunities, employment, and productivity growth, which subsequently lowers the demand for loans, and hence bank profitability. This event is also known as the delay effect. As a result of these effects, aggregate demand for bank loans decreases, putting downward pressure on lending rates [26,27]. Few studies have also suggested that low-interest rate spread causes low banking profitability in some cases. The high risk associated with the economic policy uncertainty causes a low-interest rate spread [28].

In contrast to the above explanation that economic policy uncertainty reduces interest rates, Ashraf and Shen [29] have a contradictory view. They concluded that banks increase their lending rates during policy uncertainty to cover up the additional cost associated with the risk of default loans. Boumparis et al. [30] supported the above explanation and concluded that banks increase their loan cost to cover the credit risk and lower profitability associated with the economic policy uncertainty risk. In continuation to the above studies, another strand of literature focuses on the significance of information asymmetry and banking riskiness. Previous studies highlight that economic policy uncertainty leads to information asymmetry, which influences the borrower and bank relationship. Information

asymmetry defines herding behavior and asserts that market players tend to act more like a herd when there is more vagueness regarding the accuracy of information under policy uncertainty. Ng et al. [31] suggested that sometimes because of the lack of information availability, banks are unable to evaluate the correct performance of the firms. Therefore, based on the estimates of peer banks, they make their lending decisions which sometimes lead to credit defaults. This phenomenon is known as the herding behavior of banks. Bekaert et al. [32] further added that sometimes banks purposefully make risky decisions to capture market shares during economic policy uncertainty. These decisions taken by underestimating risks and uncertainties lead to low efficiency and profitability [33].

Based on the above literature review, we can broadly infer that the economic policy uncertainty may impact bank riskiness in three ways: First, through economic meditation, high economic uncertainties generate economic disturbances, and these micro and macro uncertainties create a spillover effect on banking stability and efficiency [34]. Second, the economic policy uncertainty decreases bank earning capacity by stalling long-term investment projects, which eventually reduces credit growth and thus influences bank profitability [35]. Third, higher uncertainties create information asymmetry, which generates ambiguity in measuring firm performances, and leads to herding behavior among banks [33]. Keeping in mind these conceptual explanations and literature review, we have tried to investigate the influence of economic policy uncertainty on the Indian banking sector stability.

Besides the above literature review on the nexus between EPU and banking stability, we have also tried to draw some references between EPU and the growth in digital financial services. Previous empirical studies have concluded that the expansion of digital financial services helps in improving banking stability [36]. Financial innovation through digital financial services helps in diversifying risk, and it also helps in speeding up the process of financial inclusion [37]. Shaughnessy [38] supported the above arguments; besides, he added that the advancement in financial inclusion improves banking efficiency by increasing the credit availability and distribution cycle of banks. However, there is no conclusive evidence on the linkage between EPU and digital financial services. Moreover, considering the influence of EPU on banking and economic variables, it is evident that we can establish some relationship between EPU and the growth of digital financial services. The progress in digital financial services promotes financial intermediation, financial inclusion, and provides sustainable financial products and innovation to the unbanked population. However, EPU fosters financial sector instability and creates apprehension in the mind of the unbanked population to become a part of the formal financial system [39]. Considering this indirect link, we have also tried to investigate how volatility in economic policies influences the growth of digital financial services.

Furthermore, in addition to studying the impact of economic policy uncertainty on banking stability and digital financial services, we have also examined the influence of institutional regulation and corruption level on banking stability and digital financial services. Although as far as we understand, there is no specific study on the relationship between institutional regulation, corruption level, and the growth of digital financial services, a few recent studies have highlighted the significance of institutional regulation and corruption level on the banking stability of developed countries [40,41]. Fazio et al. [42] advocated that countries with sound institutional regulation and low corruption levels execute the government policies more efficiently compared to the countries with weak institutional regulation and high corruption levels. Institutional regulation helps in reducing transaction cost; besides, it also promotes stability with low regulatory measures. On the other hand, few studies highlighted that economic policy uncertainty increases business risk, affects external financing cost, and lowers banking profitability [43]. Therefore, banks search for alternative resources to mitigate the negative consequences of economic policy uncertainty. Against this backdrop, we have investigated how institutional regulation and corruption levels in an emerging country impact banking stability and digital financial services.

## 2.2. Theoretical and Conceptual Framework

Theoretically, there are three views on the potential impact of EPU on banking riskiness. The first theoretical concept is "real options", propounded by Bernanke (1983), which illustrates how uncertainty and the irreversibility of investment would increase the value of the option to wait, causing firms to cut or postpone investment projects. In a similar vein, EPU has the potential to drastically increase risk premiums in various financial markets, rising borrowing costs, disrupting productivity, lowering employment, and ultimately hurting the entire economy. A plethora of studies based on such theoretical concept of investment irreversibility demonstrate that uncertainty increases the value of the option to wait and that businesses can avert sunk costs by eliminating or delaying credit risks and incorporating more conventional policies [44,45]. Banks reduce their credit distribution during uncertain events and wait for the uncertainty to ease down. These decisions hamper the earning capacity of the banks, and thus profitability falls. Second, according to precautionary saving theory, firms prefer to keep extra cash in response to financial difficulty induced by cash flow unpredictability. Economic policy uncertainty, as one primary factor of uncertainty, would boost the motive for cautionary saving, encouraging enterprises to move their cash holdings upward. This event eventually increases the default rates, and hence banking asset value falls [46]. Third, another strand of theories argues that during economic policy uncertainty, due to the risk exposure of banks, depositors start demanding more interest rates. Whereas when lack of new investment reduces the demand for loans, the interest rate falls. This low profitability and earning pressure compel banks to make risky decisions, and hence chances of banking failure are high during economic policy uncertainty [47]. Using the above theoretical explanation and research gap, we proceed with our empirical investigation of exploring the nexus of economic policy uncertainty and banking stability.

## 3. Data, Variables, and Methodology

### 3.1. Data and Variables

We have employed the following proxies to measure the influence of economic policy uncertainty, institutional regulation, corruption level on banking stability, and digital financial services in India. We use the composite EPU indices developed by Baker et al. [4] to measure the economic policy uncertainty. Similarly, to measure the corruption level of an economy, we have used the corruption index of the International Country Risk Guide (ICRG). We have taken the following corruption index because of its added advantage over the Corruption Perception Index (CPI) of Transparency International and the Control of Corruption Index (CCI) of the World Bank [48]. To measure institutional regulation, we have extracted the indices of institutional regulation released by the World Bank's World Governance Indicators (WGI). We have used Z-score and NPLs as the two prominent proxies to measure banking sector stability [49]. Here, NPLs denote the nonperforming loan's percentage of total gross loans, and Z-score is the probability of default of a country's banking system, calculated as a weighted average of the Z-scores of a country's individual banks (the weights are based on the individual banks' total assets). Z-score compares a bank's buffers (capitalization and returns) with the volatility of those returns. It is estimated as (ROA+(equity/assets))/Standard Deviation (ROA). In most of the previous studies, Z-score is taken as a proxy to measure banking stability. However, to increase the robustness of our estimation, we have also included NPLs in our empirical analysis. Mobile money transaction as a percentage of GDP is used as a proxy to measure the growth of digital financial services. We have used mobile money transaction percentage of GDP because it is one of the prominent and widely used indicators to measure the progress of digital financial services [48]. In addition, based on the previous literature, we have also included specific control macroeconomic and banking variables in our empirical analysis. Moreover, due to the lack of data availability, we have extracted the annual data from 2004 to 2019. Table 1 shows in detail the sources and abbreviations of all the explanatory and outcome variables.

**Table 1.** Variables' description and data sources.

| Variable (Abbreviation) | Source |
| --- | --- |
| **Dependent Variables:** | |
| Banking stability (NPLs and Z-score) | IMF financial statistic database |
| Mobile money transaction percentage of GDP (MMT) | Financial Access Survey |
| **Independent Variables:** | |
| Economic policy uncertainty (EPU) | Baker et al. (2016) Policyuncertainty.com |
| Institutional regulation (IR) | World governance indicator |
| Corruption Index (COR) | International Country Risk Guide |
| **Control Variables:** | |
| Gross Domestic Product (GDP) | World Development indicators |
| Inflation (INF) | World Development indicators |
| Return on Assets percent of GDP (ROA) | IMF financial statistic database |
| Non-interest income to total income (NII) | IMF financial statistic database |
| (Annual data 2004–2019) | |

### 3.2. Model Specification

To estimate the relationship between EPU, institutional regulation, corruption level, banking stability, and digital financial services, we have employed the nonlinear autoregressive distribution lag estimation techniques (NARDL) suggested by Shin et al. [50]. We have used the NARDL estimation model in our empirical analysis because of its added advantages over other estimation methods. The conventional linear approaches such as the vector autoregressive approach (VAR), Engle and Granger approach, Johansen and Juselius cointegration, and ARDL bounds testing involve constant disturbance terms with constant velocity of adjustments. The postulations do not suit the financial markets [51]. It is, in essence, a dynamic error correction representation, which provides robust empirical results even for small sample sizes. Moreover, this technique also provides robust estimates in the presence of a mixed level of integration, i.e., I(0), I(1). In addition, it also checks short-run and long-run asymmetries in the data. Besides, it estimates the presence of joint and hidden cointegration [52]. The above estimating technique has one important drawback: it is inefficient when the variables are of the second order of integration (2) [53]. Therefore, we have used the Augmented Dicky Fuller unit root test to estimate the integration level among the variables. Previous studies have suggested that the traditional unit root test does not capture the presence of a structural break in the series [54]. Hence, we have also used the robust Zivot and Andrews [55] structural break test, which helps in estimating the presence of structural breaks and stationarity of the data set. Moreover, structural break indicated the presence of nonlinearity in the model. Thus, we have also used Brock, Scheinkman, Dechert, and LeBaron's [56] BDS test for confirming the presence of nonlinear dependence in the series. Shin et al. [50] recommended that both ARDL and NARDL methods use a similar estimation methodology. However, NARDL is an extension of the ARDL methods. Therefore, we have used the basic form of the unrestricted error correction linear ARDL model.

$$\Delta y_{t=\alpha} + \sum_{i=1}^{r-1} b_i \, \Delta y_{t-i+} \, \sum_{i=1}^{s-1} c_i \, \Delta x^t - i + \rho y_{t-1} + \theta x_t - 1 + \varepsilon_{t-1} \qquad (1)$$

Here, $y_t$ depicts the dependent variables that are non-performing loans (NPLs), Z-score, and mobile money transactions (MMT); $x_t$ highlights the vector of regressors; $\alpha$ represent intercept; the short-run coefficients are presented by $b_i$, $c_i$; r and s are the restricted lags and $\varepsilon_t$ is the error term. The above Equation (1) only exhibits linear response;

therefore, to estimate the short-run and long-run nonlinear response, we have borrowed the following model from Shin et al. [50].

$$y_t = \beta^+ x_t^+ + \beta^- x_t^- + \varepsilon_t. \tag{2}$$

where $\beta^+$ and $\beta^-$ represents the long-run asymmetric coefficients, $\varepsilon_i$ and $x_t$ shows error term and vector regressors, respectively. We further combine Equations (1) and (2) to derive asymmetric error correction mode which is given below:

$$\Delta y_t = a_0 \sum_{i=1}^{r-1} b_i \, \Delta y_{t-i+} \, \sum_{i=1}^{s-1}(c_i^+ \Delta x_{t-i}^+ + c_i^- \, \Delta x_{t-i}^-) + \rho y_{t-1} + \theta^+ x_{t-1}^+ + \theta^- x_{t-1}^- + e_t \tag{3}$$

where $\theta^+ = -\rho\beta^+$ and $\theta^- = -\rho\beta^-$ are a short-run adjustment for positive and negative shocks.

Previous studies have highlighted that the nonlinear ARDL model is estimated similarly to the ARDL model. Under this approach, first, we use least square regression to evaluate the error correction model, then the long-run asymmetric relationship is estimated using the bound test. Moreover, similar to ARDL analysis, in the NARDL estimation technique, we also evaluate the cointegration relationship. We use the upper, lower, and F-values to estimate the cointegration. Finally, we use the Wald test to examine the long-run and short-run symmetric and asymmetric effects on the exogenous variables. In our empirical analysis, we have used the following three NARDL models to estimate the influence of EPU, corruption, and institutional regulation on banking stability and digital financial services.

$$\Delta NPL = a_0 + \sum_{i=1}^{r-1} b_i \, \Delta NPL_{t-1} + \sum_{i=0}^{s1} c_{1,i}^+ \Delta EPU_{t-1}^+ + \sum_{i=0}^{s2} c_{1,i}^- \Delta EPU_{t-1}^- + \sum_{i=0}^{s3} c_{2,i}^+ \Delta IR_{t-1}^+ + \sum_{i=0}^{s4} c_{2,i}^- \Delta IR_{t-1}^- + \sum_{i=0}^{s5} c_{3,i}^+ \Delta COR_{t-1}^+ +$$
$$\sum_{i=0}^{s6} c_{3,i}^- \Delta COR_{t-1}^- + \sum_{i=0}^{s7} c_{4,i} \Delta GDP_{t-1} + \sum_{i=0}^{s8} c_{5,i} \Delta INF_{t-1} + \sum_{i=0}^{s9} c_{6,i} \Delta ROA_{t-1} + \sum_{i=0}^{s10} c_{7,i} \Delta NII_{t-1} + \rho NPL_{t-1} + \theta_1^+ EPU_{t-1}^+ \tag{4}$$
$$+\theta_1^- EPU_{t-1}^- + \theta_2^+ IR_{t-1}^+ + \theta_2^- IR_{t-1}^- + \theta_3^+ COR_{t-1}^+ + \theta_3^- COR_{t-1}^- + \theta_4 GDP_{t-1} + \theta_5 INF_{t-1} + \theta_6 ROA_{t-1} + \theta_7 NII_{t-1} + \varepsilon_t$$

$$\Delta LnZScore = a_0 + \sum_{i=1}^{r-1} b_i \, \Delta Zscore_{t-1} + \sum_{i=0}^{s1} c_{1,i}^+ \Delta EPU_{t-1}^+ + \sum_{i=0}^{s2} c_{1,i}^- \Delta EPU_{t-1}^- + \sum_{i=0}^{s3} c_{2,i}^+ \Delta IR_{t-1}^+ + \sum_{i=0}^{s4} c_{2,i}^- \Delta IR_{t-1}^- +$$
$$\sum_{i=0}^{s5} c_{3,i}^+ \Delta COR_{t-1}^+ + \sum_{i=0}^{s6} c_{3,i}^- \Delta COR_{t-1}^- + \sum_{i=0}^{s7} c_{4,i} \Delta GDP_{t-1} +$$
$$\sum_{i=0}^{s8} c_{5,i} \Delta INF_{t-1} + \sum_{i=0}^{s9} c_{6,i} \Delta ROA_{t-1} + \sum_{i=0}^{s10} c_{7,i} \Delta NII_{t-1} + \rho Zscore_{t-1} + \theta_1^+ EPU_{t-1}^+ + \theta_1^- EPU_{t-1}^- + \theta_2^+ IR_{t-1}^+ + \theta_2^- IR_{t-1}^- + \tag{5}$$
$$\theta_3^+ COR_{t-1}^+ + \theta_3^- COR_{t-1}^- + \theta_4 GDP_{t-1} + \theta_5 INF_{t-1} + \theta_6 ROA_{t-1} + \theta_7 NII_{t-1} + \varepsilon_t$$

$$\Delta LnMMT = a_0 + \sum_{i=1}^{r-1} b_i \, \Delta MMT_{t-1} + \sum_{i=0}^{s1} c_{1,i}^+ \Delta EPU_{t-1}^+ + \sum_{i=0}^{s2} c_{1,i}^- \Delta EPU_{t-1}^- + \sum_{i=0}^{s3} c_{2,i}^+ \Delta IR_{t-1}^+ + \sum_{i=0}^{s4} c_{2,i}^- \Delta IR_{t-1}^- +$$
$$\sum_{i=0}^{s5} c_{3,i}^+ \Delta COR_{t-1}^+ + \sum_{i=0}^{s6} c_{3,i}^- \Delta COR_{t-1}^- + \sum_{i=0}^{s7} c_{4,i} \Delta GDP_{t-1} +$$
$$\sum_{i=0}^{s8} c_{5,i} \Delta INF_{t-1} + \sum_{i=0}^{s9} c_{6,i} \Delta ROA_{t-1} + \sum_{i=0}^{s10} c_{7,i} \Delta NII_{t-1} + \rho MMT_{t-1} + \theta_1^+ EPU_{t-1}^+ + \theta_1^- EPU_{t-1}^- + \theta_2^+ IR_{t-1}^+ + \theta_2^- IR_{t-1}^- + \tag{6}$$
$$\theta_3^+ COR_{t-1}^+ + \theta_3^- COR_{t-1}^- + \theta_4 GDP_{t-1} + \theta_5 INF_{t-1} + \theta_6 ROA_{t-1} + \theta_7 NII_{t-1} + \varepsilon_t$$

Equation (4) shows the influence of explanatory variables on NPLs, Equation (5) depicts the relationship of explanatory variables with Z-score, and finally, Equation (6) highlights the impact of explanatory variables on digital financial services. All the explanatory and outcome variables are expressed in the natural logarithm form, and + and − are independent variables' positive and negative variation (IVs).

## 4. Empirical Analysis and Discussion

Table 2 exhibits the descriptive properties of all the explanatory and the outcome variables. The descriptive statistics show that the mean of NPLs in India is 6.13 percent which is more than the world average of 4 percent. The mean value of NPLs of a few developed and emerging countries such as the U.S. (3.09), China (1.83), Indonesia (4.09), UK (4.59), and South Africa (3.57) is also less than the average of India. However, the mean of NPLs of India is well off compared to the cumulative average of MENA countries (7.6 percent) and the sub-Saharan Africa region (7.63 percent). In context to Z-score, the mean of India is 15.95, which is satisfactory compared to sub-Saharan Africa (7.7 percent), Latin America and the Caribbean (13 percent), and Europe and Central Asia (6.31 percent). The descriptive statistic shows that the mean value of the EPU of India is 105.77, which is comparatively lower than most of the regions. For instance, the mean value of EPU

in Asia is 107, in the European Union it is 119, in non-European Union countries it is 112, and in Europe it is 119 [57]. However, during the last few years, the average EPU of India has increased from the previous average of 97. The mean value of mobile money transactions in India (1.8) is lesser than the mean value of developed countries. For instance, in some countries such as China (309.01) and Russia (404.81), the growth in the percentage of mobile and internet-based transactions is exorbitant. Similarly, the mean of institutional regulation (−0.29) is low in comparison to developed countries such as the U.S. (1.35), the UK (1.63), and Japan (1.33). According to the World Bank report, 2019, India ranks 92 in terms of institutional regulation out of 192 countries. In addition to the above variables, descriptive statistics also conclude that corruption level (2.4), growth rate (6.88), inflation (10.66), and percentage of non-interest income (30.04) are moderate in India compared to other emerging and developing regions globally. Moreover, the probability value of the Jarque–Bera test confirms that the data set is not normal, which further strengthens the reason for applying asymmetric methodologies.

**Table 2.** Descriptive statistics.

| Variables | Mean | Median | Maximum | Minimum | Std.Dev | Jarque–Bera | Probability |
|---|---|---|---|---|---|---|---|
| NPLs | 6.13 | 5.11 | 9.98 | 3.37 | 3.06 | 1.251 | 0.6100 |
| Z-score | 15.95 | 17.01 | 17.28 | 16.64 | 0.26 | 0.342 | 0.0632 |
| MMT | 1.88 | 0.46 | 7.46 | 0.018 | 2.99 | 3.245 | 0.1901 |
| EPU | 105.47 | 85.32 | 185.46 | 70.89 | 45.82 | 2.525 | 0.4201 |
| IR | −0.29 | −0.27 | −0.16 | −0.44 | 0.10 | 1.342 | 0.6110 |
| COR | 2.4 | 2.50 | 2.61 | 2.00 | 0.20 | 3.667 | 0.0031 |
| GDP | 6.88 | 6.89 | 8.16 | 5.46 | 0.93 | 1.354 | 0.0723 |
| INF | 10.66 | 10.80 | 13.90 | 7.20 | 2.90 | 2.453 | 0.1625 |
| ROA | 0.91 | 0.88 | 1.38 | 0.48 | 0.41 | 0.352 | 0.0101 |
| NII | 30.04 | 28.08 | 35.64 | 26.62 | 3.98 | 2.346 | 0.0001 |

*4.1. Unit Root Test*

One of the preconditions for applying the NARDL analysis is that none of the variables have to be integrated at the second level I(2). Therefore, to confirm the level of integration, we have used the Augmented Dicky Fuller test (ADF). Table 3 presents the results of the ADF test, which concludes that, except for non-performing loans, corruption level, and inflation which are stationary at levels, all the other variables are stationary at first difference. The mixed level of integration further encourages us to proceed with the NARDL estimation.

Previous literature has concluded that conventional unit root does not consider structural breaks in the data sets resulting in misleading results. Therefore, to substantiate the results of the ADF test and to measure the presence of structural breaks in the data sets, we have employed the Z&A unit root test. Table 4 exhibits the results of the Z&A unit root test. The test confirms the presence of a structural break in the data set; besides, the result also concludes that the years 2008 to 2010 are the frequent structural break in most of the variables because these years experienced the highest uncertainty and volatilities due to the post-global financial recession. The data set of digital payments has witnessed structural breaks during the years 2012 and 2017. Because there was a surge in foreign investments in India in 2012, this has resulted in the development of digital infrastructure in India. Similarly, in 2016, India participated in a joint conclave on digital growth, which has assisted in framing appropriate measures to increase digital transactions. Further, the

Z&A structural break test confirms that none of the variables are stationary at the second difference I(2).

**Table 3.** Augmented Dicky Fuller unit root test result.

| Variables | Levels | | First Difference | |
| --- | --- | --- | --- | --- |
| | Constant (at 5 Percent) | Constant and Trends (at 5 Percent) | Constant (at 5 Percent) | Constant and Trends (at 5 Percent) |
| NPLs | −1.09 (−1.043) * | −1.19 (−2.713) | −3.59 (−1.208) | −4.19 (−2.137) |
| Z-score | −2.18 (−2.012) | −2.91 (−2.012) | −2.91 (−2.746) * | −2.98 (−3.045) |
| MMT | −1.28 (−1.142) | −1.85 (−2.817) | −2.21 (−2.429) ** | −3.01 (−2.178) |
| EPU | 1.18 (−1.081) | −2.63 (−2.095) * | −2.87 (−1.853) | −3.12 (−2.409) * |
| IR | −2.03 (−2.291) | −2.32 (−2.843) | −3.09 (−2.417) * | −3.45 (−2.912) |
| COR | 1.31 (−1.837) * | −1.22 (−2.109) | −2.17 (−1.971) | −3.41 (−2.194) * |
| GDP | −2.12 (−2.018) | −2.82 (−2.071) | −2.00 (−2.116) * | −3.99 (−2.240) |
| INF | −2.15 (−2.110) * | −2.81 (−1.108) | −0.94 (−1.576) | −2.91 (−2.751) |
| ROA | −1.94 (−2.121) | −2.20 (−2.144) | −3.52 (−2.988) * | −3.82 (−2.392) |
| NII | −1.84 (−1.912) | −1.87 (−1.619) | −2.69 (−3.943) * | −2.09 (−2.328) |

*, ** at 10, 5, and 1 percent level of significance.

**Table 4.** Zivot and Andrews test.

| Variables | Levels | | First Difference | |
| --- | --- | --- | --- | --- |
| | *t*-Statistic | Time Break | *t*-Statistics | Time Break |
| NPLs | −1.742 * | 2008 | −2.1091 * | 2008 |
| Z-score | −2.1983 * | 2008 | −3.4742 * | 2010 |
| MMT | −3.5284 * | 2012, 2017 | −3.8734 *** | 2012, 2017 |
| EPU | −2.6793 | 2009 | −4.1834 * | 2010 |
| IR | −3.2464 | 2008 | −2.8231 * | 2008 |
| COR | −2.9032 | 2010 | −5.7263 * | 2010 |
| GDP | −3.2577 * | 2010 | −4.6180 ** | 2010 |
| INF | −2.4722 * | 2004 | −2.6590 * | 2005 |
| ROA | −2.1983 ** | 2009 | −4.3853 *** | 2009 |
| NII | −2.3732 * | 2008 | −3.4212 * | 2008 |

*, **, *** at 10, 5, and 1 percent level of significance.

*4.2. BDS Test*

The presence of a structural break in the data set augments the application of the BDS test, as this test helps in estimating the nonlinear dependence. Table 5 presents the results of the BDS test, which confirms that the variables are not identical and equally distributed. These results demonstrate the nonlinear properties and hence encourage to use the NARDL approach.

**Table 5.** BDS test.

| BDS Variables | Embedded Dimensions = m | | | | |
|---|---|---|---|---|---|
| | m = 2 | m = 3 | m = 4 | m = 5 | m = 6 |
| NPLs | 0.1812 ** | 0.1965 ** | 0.2122 *** | 0.2389 ** | 0.2399 ** |
| Z-score | 0.2378 ** | 0.3327 ** | 0.3764 ** | 0.3891 *** | 0.2184 ** |
| MMT | 0.1129 ** | 0.1781 *** | 0.2342 ** | 0.2843 ** | 0.3128 ** |
| EPU | 0.1992 ** | 0.2198 ** | 0.2764 ** | −0.2931 *** | 0.3185 *** |
| IR | 0.2842 ** | 0.2954 ** | 0.3175 ** | 0.3983 * | 0.3871 *** |
| COR | 0.1274 ** | 0.1883 *** | 0.2147 ** | 0.2582 *** | 0.2743 ** |
| GDP | 0.1338 *** | −0.0572 ** | 0.1454 ** | 0.1783 ** | 0.2421 ** |
| INF | 0.2182 ** | 0.1809 *** | 0.1933 *** | 0.1965 *** | 0.1997 ** |
| ROA | 0.2313 ** | 0.2753 *** | 0.3532 * | −0.3771 ** | 0.3939 *** |
| NII | 0.0572 ** | 0.3133 ** | 0.2914 *** | 0.2859 *** | 0.1742 *** |

*, **, *** at 10, 5, and 1 percent level of significance.

Subsequent to confirming the unit root and structural break, we proceed with the short-run and long-run NARDL estimation. To empirically discuss the influence of economic policy uncertainty, institutional regulation, and corruption on banking stability and digital finance, we have presented the analysis in three parts. In Table 6, we have exhibited the influence of EPU, institutional regulation, and corruption on NPLs. Subsequently, in Table 7, we have presented the analysis of EPU, institutional regulation, and corruption on Z-score. Finally, in Table 8, we have highlighted the impact of EPU, institutional regulation, corruption on digital finance. In all three models, initially, we have examined the cointegration relationship between the explanatory and outcome variables. According to Bahmani-Oskooee and Bohl [58] and Stock and Watson [59], the selection of optimum lag is binding before estimating the cointegration relationship. Therefore, considering the Akaike information criteria (AIC), several models are evaluated, and the one with lower AIC is finally selected. Next, we estimated the asymmetric cointegration among the dependent and independent variables in all three models. The cointegration results show that the F-value (6.36, 4.59, 5.89) in all three models is more than the upper bound values at a 5 percent significance level. It implies that we can reject the null hypothesis of no cointegration and accept the alternative hypothesis of cointegration. This indicates that there is cointegration among the explanatory and outcome variables in all three models. Finally, after confirming the long-run asymmetric cointegration in all three models, we proceed with estimating the long-run and short-run impact of positive and negative shocks of independent variables on the dependent variables for each model separately.

The first model presented in Table 6 shows that in the short run, positive shocks of EPU have a positive impact on the NPLs, whereas negative shocks of EPU have a negative influence on NPLs. This shows that with a 1 percent increase in EPU, NPLs increase by 0.32 percent, and with a 1 percent decrease in EPU, NPLs fall by 0.15 percent. Similarly, the analysis also concludes that in the short-run, positive shocks of corruption positively influence NPLs, whereas a negative shock in the corruption level has a negative impact on the NPLs. Further, the empirical findings substantiate that positive shocks of the institutional regulation create a negative impact on NPLs and vice versa. Based on the short-run outcomes, we can infer an asymmetric relation between EPU, institutional regulation, corruption, and NPLs. The results of the short-run Wald test also confirm an asymmetric relationship between the explanatory and outcome variables. Similar to the short-run results, the long-run result also confirms the same asymmetric relationship between EPU, institutional regulation, corruption, and NPLs. However, the magnitude is high in the long run compared to the short run. This depicts that the influence of EPU,

corruption, and institutional regulation is more in the long run. For instance, with a 1 percent increase in the EPU in the long run, NPL increases by 1.48 percent in India.

**Table 6.** NARDL result.

| NARDL Short-Run Result | Lags | | |
|---|---|---|---|
| Dependent Variable: NPLs | 0 | 1 | 2 |
| $\Delta$ Ln EPU$^+$ | 0.12 (0.32) ** | 0.18 (0.07) | 0.32 (1.95) |
| $\Delta$ Ln EPU$^-$ | −0.56 (−0.15) * | 0.31 (0.82) | −0.62 (−0.43) |
| $\Delta$ Ln IR$^+$ | 0.21 (0.51) | −0.11 (−0.12) * | 0.31 (0.11) |
| $\Delta$ Ln IR$^-$ | 0.63 (0.11) | 0.16 (0.91) * | 0.33 (1.14) |
| $\Delta$ Ln COR$^+$ | 0.06 (1.75) | 0.51 (0.76) * | 0.41 (2.84) |
| $\Delta$ Ln COR$^-$ | 0.17 (0.39) | −0.44 (−0.43) ** | 0.65 (1.19) |
| $\Delta$ Ln GDP | −0.23 (−0.49) | 0.22 (0.18) | −0.65 (−0.83) |
| $\Delta$ Ln INF | 0.24 (0.31) ** | 0.09 (0.23) | 0.19 (1.19) |
| $\Delta$ Ln ROA | 0.84 (0.14) | 0.18 (1.02) | 0.54 (0.89) |
| $\Delta$ Ln NII | 0.11 (0.34) | 0.41 (1.23) | 0.06 (1.19) |
| NARDL Long-Run Result | | | |
| Ln EPU$^-$ | Ln EPU$^+$ | Ln COR$^-$ | Ln COR$^+$ | Ln IR$^-$ |
| −0.10 (−1.01) ** | 0.03 (1.48) * | −0.18 (−1.15) * | 0.17 (1.67) ** | 0.19 (1.09) * |
| Ln IR$^+$ | Ln GDP | Ln INF | Ln ROA | Ln NII |
| −0.09 (−1.14) * | −0.12 (−1.02) * | 0.25 (1.57) ** | −1.23 (−0.53) * | −0.23 (−1.12) ** |
| Diagnostic Test Results: | | | |
| ECMt−1 | (Joint Sig) | Adj. R2 | RESET | LM |
| −0.010 (0.00 ***) | 8.16 *** | 0.61 | 4.091 (0.512) | 0.78 (0.452) |
| F Statistic | Ln EPUSR | Ln EPULR | Ln CORSR | Ln CORLR |
| 6.36 | 0.03 (0.002) * | 2.31 (0.05) * | 0.76 (0.01) | 1.12 (0.002) * |
| Ln IRSR | Ln IRLR | | |
| 1.09 (0.005) | 1.22 (0.000) ** | | |

*, **, *** indicate the rejection of the null hypothesis at 10, 5, and 1 percent level of significance. LR: long-run Wald result, SR: short-run Wald result. Lower bound and upper bound value at 5 percent level of significance, 1.56 and 3.15.

Furthermore, model 1 also states, in context to control variables, in the long run, economic growth rate, banking profitability, and non-interest income exert a negative impact on the NPLs. However, inflation has a positive influence on NPLs in the long run. It indicates that in the long-run, economic growth, profitability, and other income sources help reduce NPLs, whereas inflation increases NPLs. However, we also need to be cautious about the reverse causality, where higher NPLs may reduce banking profitability and liquidity. Therefore, to maintain stable banking, adequate measures need to be undertaken to maintain a balance between profitability and bad loans. The long-run Wald estimates also confirm an asymmetric relationship between EPU, institutional regulation, corruption, and NPLs. The negative and significant value of the error correction model further strengthens the results of Wald estimates. In addition to the above short-run and long-run findings, we have also performed several diagnostic tests to support the viability of our estimates. The r-square value of 0.61 indicates that the model is stable and appropriate. Similarly, the Lagrange multiplier and Ramsey RESET test confirm no serial correlation, and the model is well specified. In addition, the nonlinearity curve (Figure A1) also substantiates the asymmetric relationship between EPU and NPLs in the long run and confirms that the influence of positive shocks of EPU is more powerful than the negative shocks.

**Table 7.** NARDL result.

| NARDL Short-Run Result | Lags | | |
|---|---|---|---|
| **Dependent Variable: Z-Score** | **0** | **1** | **2** |
| $\Delta$ Ln EPU$^+$ | 0.19 (0.02) | −0.04 (−0.63) ** | 0.18 (0.67) |
| $\Delta$ Ln EPU$^-$ | 0. 45 (1.05) | 0.11 (0.91) ** | 1.16 (0.23) |
| $\Delta$ Ln IR$^+$ | 0.04 (0.11) * | 0.09 (0.35) | 0.16 (0.18) |
| $\Delta$ Ln IR$^-$ | −0.23 (−0.43) * | 1.21 (0.27) | 0.62 (0.09) |
| $\Delta$ Ln COR$^+$ | −0.13 (−1.19) * | 0.43 (1.03) | 0.26 (1.33) |
| $\Delta$ Ln COR$^-$ | 0.17 (1.03) * | 0.12 (1.84) | 0.10 (1.14) |
| $\Delta$ Ln GDP | 0.11 (1.31) * | 1.22 (1.08) | 0.21 (0.38) |
| $\Delta$ Ln INF | −0.39 (−0.17) * | 0.08 (0.49) | 0.41 (1.07) |
| $\Delta$ Ln ROA | 0.14 (0.18) | 0.22 (0.17) * | 0.35 (0.61) |
| $\Delta$ Ln NII | 0.10 (0.37) | 0.22 (1.16) * | 0.13 (1.73) |
| NARDL Long-Run Result | | | |
| Ln EPU$^-$ | Ln EPU$^+$ | Ln COR$^-$ | Ln COR$^+$ | Ln IR$^-$ |
| 0.11 (0.82) * | −0.37 (−1.12) ** | 0.62 (0.15) * | −0.25 (−1.54) * | −0.38 (−1.31) * |
| Ln IR$^+$ | Ln GDP | Ln INF | Ln ROA | Ln NII |
| 0.16 (1.27) * | 0.16 (1.54) * | 0.19 (1.02) | 1.15 (0.28)* | 0.15 (0.75) * |
| **Diagnostic Test Results:** | | | | |
| ECMt-1 | (Joint Sig) | Adj. R$^2$ | RESET | LM |
| −0.009 (0.00 **) | 6.16 ** | 0.68 | 6.09 | 14 |
| F | Ln EPU$_{SR}$ | Ln EPU$_{LR}$ | Ln COR$_{SR}$ | Ln COR$_{LR}$ |
| 4.59 | 0.06 (0.00) * | 1.01 (0.03) * | 0.36 (0.00) | 1.03 (0.00) * |
| Ln IR$_{SR}$ | Ln IR$_{LR}$ | | | |
| 1.00 (0.01) * | 1.09 (0.00) * | | | |

*, ** indicate the rejection of the null hypothesis at 10, 5, and 1 percent level of significance. LR: long-run Wald result, SR: short-run Wald result. Lower bound and upper bound value at 5 percent level of significance, 2.04 and 3.87.

After confirming the asymmetric relationship between EPU, institutional regulation, corruption, and NPLs, we proceed with estimating the influence of independent variables on banking Z-score. The Z-score is used as a second proxy to measure the Indian banking sector's stability. Table 7 reports the findings of the second model. The result shows that a positive shock of EPU increases banking sector instability in the short run. On the contrary, a negative shock of EPU assists in reducing banking sector instability. This depicts that a 1 percent increase in EPU increases banking sector instability by 0.63 percent, whereas a 1 percent decrease in EPU increases banking stability by 0.91 percent. Similar to the above findings, the short-run result also depicts that a positive corruption shock positively influences the banking sector instability, whereas a negative shock of corruption reduces banking instability. This suggests that a 1 percent increase in corruption increases banking instability by 1.19 percent. Besides, model 2 also concludes that a positive shock of institutional regulation increases banking stability, contrary to a negative shock, which reduces banking stability. In addition to the independent variables, the short-run result also highlights that positive economic growth and higher non-interest income helps in increasing banking sector stability. However, rampant inflation negatively influences banking sector stability. Based on the empirical findings, we can conclude an asymmetric relationship between the explanatory and outcome variable in the short run. The short-run Wald test also strengthens the above empirical findings.

**Table 8.** NARDL result.

| NARDL Short-Run Result | Lags | | |
|---|---|---|---|
| **Dependent Variable: MMT** | **0** | **1** | **2** |
| Δ Ln EPU$^+$ | −0.18 (−0.51) ** | 1.11 (0.06) | 0.12 (1.45) |
| Δ Ln EPU$^-$ | 0.23 (0.44) * | 0.38 (0.64) | 0.29 (0.21) |
| Δ Ln IR$^+$ | 0.05 (0.11) | 0.09 (0.48) | 0.48 (1.09) |
| Δ Ln IR$^-$ | 0.74 (1.54) | 1.14 (0.70) | 0.27 (1.39) |
| Δ Ln COR$^+$ | 0.27 (1.25) | −0.32 (−0.30) ** | 1.19 (1.04) |
| Δ Ln COR$^-$ | 0.21 (0.62) | 0.35 (1.03) * | 0.28 (2.29) |
| Δ Ln GDP | 0.56 (0.28) * | 0.19 (1.18) | 0.70 (0.39) |
| Δ Ln INF | 0.36 (0.53) | 0.17 (0.03) | 0.47 (1.52) |
| Δ Ln ROA | 0.09 (0.02) * | 0.37 (1.63) | 0.65 (0.56) |
| Δ Ln NII | 0.48 (1.03) | 0.26 (1.02) | 0.39 (1.41) |
| NARDL Long-Run Result | | | |
| Ln EPU$^-$ | Ln EPU$^+$ | Ln COR$^-$ | Ln COR$^+$ | Ln IR$^-$ |
| 0.51 (1.01) * | −0.24 (−1.23) ** | 0.12 (0.36) * | −0.47 (−2.04) ** | −0.19 (−0.37) * |
| Ln IR$^+$ | Ln GDP | Ln INF | Ln ROA | Ln NII |
| 0.28 (0.87) * | 0.20 (1.12)* | 0.19 (0.27) | 1.29 (0.31) * | 0.28 (0.49) * |
| **Diagnostic Test** | | | | |
| ECMt-1 | (Joint Sig) | Adj. R$^2$ | RESET | LM |
| −0.007 (0.00 **) | 9.21 *** | 0.64 | 6.11 | 13 |
| F | Ln EPU$_{SR}$ | Ln EPU$_{LR}$ | Ln COR$_{SR}$ | Ln COR$_{LR}$ |
| 5.81 | 1.01 (0.000) * | 1.87 (0.01) * | 0.18 (0.002) * | 1.07 (0.00) * |
| Ln IR$_{SR}$ | Ln IR$_{LR}$ | | | |
| 1.00 (0.36) | 1.02 (0.00) * | | | |

*, **, *** indicate the rejection of the null hypothesis at 10, 5, and 1 percent level of significance. LR: long-run Wald result, SR: short-run Wald result. Lower bound and upper bound value at 5 percent level of significance, 2.12 and 3.78.

The long-run results of model 2 also corroborate with the short-run findings. The long-run results show that the positive shock of EPU exerts a negative impact on banking stability. It implies that higher economic policy uncertainty results in low banking instability. The magnitude of the long run is higher (−1.12) than the short-run (−0.63) findings. This suggests that the impact of EPU is severe in the long run compared to the short run. The long-run findings also highlight that negative (positive) shocks of corruption (institutional regulation) assist in maintaining banking sector instability. In context to the above variables, long-run estimates also exhibit that economic growth, banking profitability, and non-interest income helps in strengthening banking sector stability in the long run. The long-run Wald test estimates are consistent with the long-run estimations. Like model 1, we have also performed several diagnostic tests to confirm the robustness of model 2. The R2 value of 0.68 supports model fitness. The value of the LM and Ramsey RESET test also verifies that the model is correctly specified and there is no serial correlation. Finally, the negative and significant value of the error correction model and the asymmetric curve of model 2 (Figure A2) also supports our empirical estimates and it also strengthens the argument that the positive influence of EPU is more influential than the negative shocks. The long-run outcome is in line with the studies of Nguyen [13] and Bilgin et al. [14].

Conclusively, we can infer from models 1 and 2 that EPU, corruption, and institutional regulation have an asymmetric impact on the Indian banking sector's stability. An increase

in policy uncertainty significantly influences banking sector NPLs and stability. Economic policy uncertainties hampering banking stability may be due to the following reasons. First, high micro and macro uncertainties destabilize the proper functioning of the prominent industries, reducing their profitability and indirectly influencing debt servicing capabilities. Second, from the demand side, higher uncertainties reduce the earning capacity of individuals, which eventually impedes the demand of the product, and thus lower demand impacts the earning of individuals and industries. Therefore, again, the risk of default increases. Third, from the banker's perspective, higher EPU sometimes induces banks to make risky decisions, for instance, making wrong investment decisions, providing loans to individuals without proper credit verifications, making risky decisions to make quick profits, etc. These decisions further destabilize banking stability in the long run. Fourth, higher EPU reduces the demand for new capital investments in terms of banking profitability. These low investment demands further reduce industrial credit demand, and hence the comprehensive influence of these collective decisions reduces banking profitability and stability. These findings and explanations are corroborated by Karadima and Louri [60].

In addition to the above justifications on the asymmetric relationship between EPU and banking stability, the following reasons justify the relationship between corruption, institutional regulation, and banking stability. The corruption level increases banking stability in two ways, first, by increasing default loans and, second, by reducing banking profitability. Default loans may increase because sometimes borrowers use corrupt practices to take loans more than their paying capacity. Hence, due to such kinds of borrower's, the percentage of default loans increases. Sometimes, bankers use corrupt practices to approve loans for bad clients, which eventually reduces banking efficiency and profitability. These practices, in the long run, hamper banking sector instability. The studies conducted by Bougatef [61], Goel and Hasan [62], and Son et al. [63] support the above explanations. In context to institutional regulation, we can infer that a stringent regulatory framework reduces the chances of laxity in supervision. It also promotes efficiency and productivity of workers, and therefore the chances of banking stability increase. The findings also support the bad management hypothesis, which entails that a lower regulatory mechanism promotes lower efficiency and management. Hence, the chances of instability increase in the long run [64,65].

Finally, after confirming the asymmetric influence of EPU, corruption, and institutional regulation on the banking sector stability, we estimated the impact of EPU, corruption level, and institutional regulation on the growth of digital banking services. Table 8 presents the empirical findings of model 3, which portrays how EPU, corruption, and institutional regulatory framework influence mobile money and internet-based transaction in India. The NARDL short-run estimates confirm that positive (negative) shocks of EPU have a negative (positive) impact on the growth of digital financial services. It implies that an increase in economic uncertainty hampers the progress of digital financial services, whereas a decrease in policy uncertainty accelerates the progress of digitalization. Correspondingly, short-run results also validate that positive shocks of the corruption level impede the growth of digital financial services, whereas a lower corruption level boosts the digital financial services. Furthermore, the result also concludes that institutional regulation has no significant impact on the growth of digital financial services in the short run. In addition to the above findings, short-run estimates verify that growth rate and banking profitability also have a significant and positive impact on the growth of digital financial services. The short-run Wald test estimates also validate an asymmetric relationship between EPU, corruption, and digital financial services.

Like the short-run estimates, long-run estimates also validate the asymmetric influence of EPU, corruption, and institutional regulation on digital financial services. The result concludes that higher EPU lowers the growth of digital financial services. The long-run estimates also highlight that, in comparison to the short-run, the long-run magnitude of EPU is much higher. This indicates that a 1 percent increase in EPU decreases digital financial services by −1.23 percent in the long run. The long-run results also depict that

proper institutional regulations increase the growth of digital financial services, whereas laxity in regulations hampers the prospects of digital financial services. The 1 percent increase in institutional regulations increases the digital financial services by 0.87 percent in the long run. About the corruption level, the long-run results iterate that a higher corruption level decreases the growth of digital financial services and vice versa. The long-run results validate that the impact of all the three explanatory variables is higher in the long run compared to the short run. The above results restate an asymmetric relationship between EPU, corruption, institutional regulation, and digital financial services in the long run. The WALD test and the error correction model also strengthen the above estimates. Finally, we have also validated the robustness of the above model by including several diagnostic tests. We have employed the Lagrange multiplier and Ramsey RESET test to validate serial correlation and model specification. Both the tests substantiate that there is no serial correlation, and the model is well specified. The $R^2$ (0.64) value also authenticates that the model is fit, besides the NARDL asymmetric curve (Figure A3) also strengthening the asymmetric relationship between EPU and digital financial services and collaborating with the empirical outcome.

Based on the above outcome, we can state that EPU has an asymmetric influence on the growth of digital financial services. Higher EPU reduces the progress of digital financial services due to the following reasons. As already discussed, the above economic policy uncertainty reduces the stability and profitability of banks. Banks' low stability and profitability further curtail the investment in digital financial services. The pessimistic environment created by EPU also defers the bank- and government-level investment in the countries' digital infrastructure. Therefore, higher EPU in emerging countries reduces the growth of digital financial services.

In addition to the relationship between EPU and digital financial services, the current study also concludes that corruption level and institutional regulation also have an asymmetric relationship with digital financial services in the long run. The following reasons fortify the above findings. Previous studies have concluded that a higher corruption level in any country hampers investments and growth prospects. Corruption disrupts the correct channel of investment and encourages forgery and malpractices. The same explanation also applies to the banking industry. A high level of corruption reduces the profitability and efficiency of banks and thus impedes investment programs. In emerging countries such as India and Brazil, loopholes in the regulatory framework and government machinery promote corruption, hence stalling the growth of digital financial services. Besides corruption, weak institutional regulations also halt the progress of digital financial services because a weak institutional framework promotes inefficiency and leads to policy paralysis [66]. Lack of efficient supervision, planning, and policy orientation impede digital financial services. The following studies support the above explanations: Lee et al. [67], Huang et al. [68], and Syed et al. [69].

## 5. Conclusions and Policy Implications

Some recent studies have examined the adverse influence of EPU on micro and macroeconomic policies. However, the impact of EPU on banking stability, which is an essential component of economic development, is still overlooked. To bridge this gap in the current study, we have investigated the asymmetric influence of EPU, corruption, and institutional regulation on Indian banking stability and the growth of digital financial services. Using the country-level annual data of India from 2004 to 2019, we conclude the following findings: First, based on the ADF unit root test and Z&A structural break test, the study concludes structural break and mixed integration in the series. Second, the results of the bound test confirm that there is long-run cointegration between the explanatory and the outcome variables. Finally, all the models of the NARDL approach infer that there is an asymmetric relationship between EPU, corruption, institutional regulation, banking stability, and digital financial services in the long run. It implies that positive (negative) shocks of EPU, corruption level, and institutional regulation have a correspondingly positive (negative)

influence on banking stability and digital financial services. The study concludes that in the long run, a 1 percent increase in the EPU increases NPLs by 1.48 percent and decreases Z-score by −1.12 percent.

Similarly, a 1 percent increase in EPU reduces the progress of digital financial services by −1.23 percent. Our interesting findings imply that EPU is a prominent determinant of risk that impacts banking stability, NPLs, and digital financial services. The policymakers should draw appropriate plans to reduce the adverse impact of EPU on banking stability and digital financial services. For instance, they should make appropriate capital buffers to handle such policy uncertainties because the capital buffer is vital to strengthen banks' resilience to risk at higher levels of policy uncertainty. In addition, the bank should also be restraint from making any hasty decisions during policy uncertainties. Instead, they should employ the strategy of wait and watch to tackle such situations. The study also reveals that weak institutional regulation is also one of the reasons for high banking instability. Therefore, policymakers must take adequate measures to strengthen the institutional regulatory framework. We can infer from the previous literature that a robust regulatory framework helps in reducing EPU and corruption in developed countries. In addition, it also increases banking stability in such countries. Therefore, taking cognizance from the developed countries, our policymakers should also make efforts to make the institutional regulations more robust and accountable. Considering the opaque nature of bank assets, especially in times of high EPU, policymakers should enable regulations on private monitoring to embolden private investors to observe and exercise effective governance over banks. At the macro level, the government should also measure how frequently the volatility in EPU influences bank risk and adequately make provisions to reduce the spillover. In addition, in context to the control variable, adequate attention should be given to the reverse causality between ROA and NPLs, and reserve provision needs to be maintained to mitigate the adverse influence of NPLs on ROA.

Finally, the study can be replicated in the future by observing the impact of EPU on Islamic banks and conducting a comparative analysis between public and private banks. In addition, future studies can also be performed by individually analyzing the interaction analysis of EPU, digital financial services, and banking stability.

**Author Contributions:** Conceptualization, A.A.S., M.A.K., A.U. and S.G.; methodology, A.A.S., M.A.K., A.U. and S.G.; software, A.A.S., M.A.K., A.U. and S.G.; validation, A.A.S., M.A.K., A.U. and S.G.; formal analysis, A.A.S., M.A.K., A.U. and S.G.; investigation, A.A.S., M.A.K., A.U. and S.G.; resources, A.A.S., M.A.K., A.U. and S.G.; data curation, A.A.S., M.A.K., A.U. and S.G.; writing—original draft preparation, A.A.S., M.A.K. and A.U.; writing—review and editing, A.A.S., M.A.K., A.U. and S.G.; visualization, A.A.S., M.A.K. and A.U.; supervision, S.G.; project administration, A.A.S., M.A.K., A.U. and S.G. All authors have read and agreed to the published version of the manuscript.

**Funding:** This research received no external funding.

**Institutional Review Board Statement:** The study was conducted in accordance with the Declaration of Helsinki and approved by the Institutional Review Board (or Ethics Committee) of the University of Malta (protocol code FEMA-2022-00090 and 08-02-2022). Studies did not involve humans or animals.

**Informed Consent Statement:** Not applicable.

**Data Availability Statement:** The data used in the study are widely available on open directories sources such as the World Bank policy uncertainty.com.

**Conflicts of Interest:** The authors declare no conflict of interest.

## Appendix A

Dynamic Multiplier Graph.

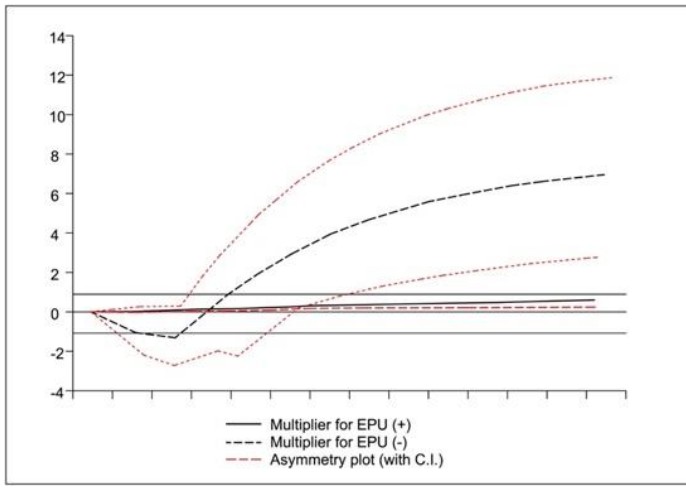

**Figure A1.** Model 1 dependent variable (NPLs).

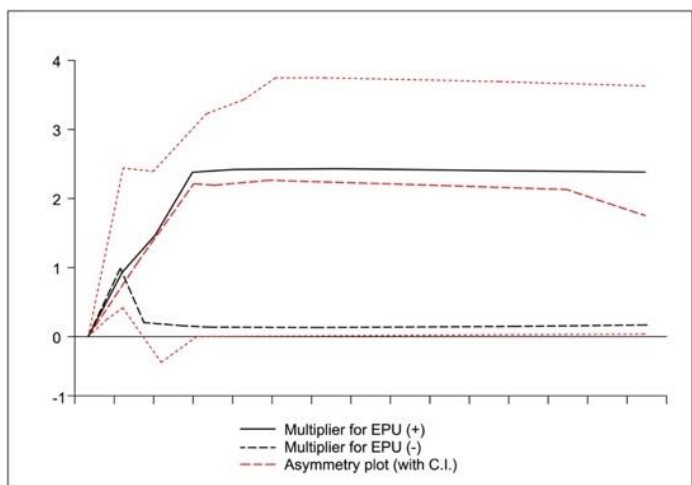

**Figure A2.** Model 2 dependent variable (Z-score).

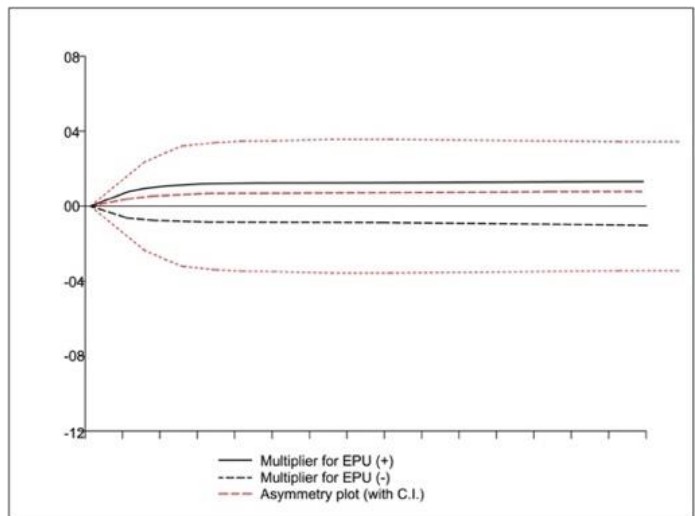

**Figure A3.** Model 3 dependent variable (MMT).

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
