# Peer review of "An Asymmetric Analysis of the Influence That Economic Policy Uncertainty, Institutional Quality, and Corruption Level Have on India’s Digital Banking Services and Banking Stability"

_sustainability, doi:10.3390/su14063238_

Round 1
Reviewer 1 Report
The novelty of the project is justified. However, neither the approach itself nor the listed elements are well-established, and therefore not only a clear description of them is required, but also a justification, first of all, of the necessity and sufficiency of their application in just such a quantity, sequence of application, interconnection, complementarity.
In this paper, authors used 59 sources, containing both historical and fundamental works, as well as the latest scientific research on this topic. But the literature review can be structured. The papers discussed many points of this study. Please, discuss these papers:
An, J., Mikhaylov, A. (2020). Russian energy projects in South Africa. Journal of Energy in Southern Africa, 31(3). http://dx.doi.org/10.17159/2413-3051/2020/v31i3a7809
Bhuiyan M.A., Dinçer H., Yüksel S., Mikhaylov A., Danish M.S.S., Pinter G., Uyeh D.D., Stepanova D. (2022). Economic indicators and bioenergy supply in developed economies: QROF-DEMATEL and random forest models. Energy Reports, 8, 2022, 561-570 https://doi.org/10.1016/j.egyr.2021.11.278
At the same time, the above reflects the instrumental aspects, but the proposed tools are aimed at application, and therefore it is necessary to justify such an application to the selected object with the identification of its advantages in comparison with other methods used. In the absence of such information, the scientific novelty of the project seems unreasonable. Thus, the proposed tools are aimed at application, and therefore it is necessary to justify such an application to the selected object with the identification of its advantages in comparison with other methods used.
The comments presented above regarding novelty are valid for the analysis of the current state, therefore it seems that for a deep analysis of the current state, these comments should be eliminated. It is not clear how the effectiveness of the proposed method will be determined.
Authors need to add more details on the range of simulation considered in this work should be clearly outlined within the abstract and in Table 1 and Figure 1. The current statements are vague and too general to get an idea of the work that have been accomplished.
Author Response
Dear Reviewer
Thank you for the kind suggestions and comments which has made our paper flow better.
Kindly see the attachment

Reviewer 2 Report
Main comments are reported in the pdf.

Author Response
Dear reviewer
Thank you for the comments and suggestions which has made our paper flow much better.
Please find attached the replies.
Thank you
Kind regards

Round 2
Reviewer 2 Report
I feel the contribution has improved and I think overall it is worth publishing for the originality has been better highlighted.